# Watermarking LLMs with Weight Quantization

**Linyang Li**[*123], **Botian Jiang**[12*] **Pengyu Wang**[12], **Ke Ren**[12],
**Hang Yan**[3], **Xipeng Qiu**[12 †]

[1]School of Computer Science, Fudan University
[2]Shanghai Key Laboratory of Intelligent Information Processing, Fudan University
[3]Shanghai AI Laboratory
{btjiang23, pywang22, kren22}@m.fudan.edu.cn
yanhang@pjlab.org.cn
{linyangli19, xpqiu}@fudan.edu.cn

## Abstract

Abuse of large language models reveals high risks as large language models are being deployed at an astonishing speed. It is important to protect the model weights to avoid malicious usage that violates licenses of open-source large language models. This paper proposes a novel watermarking strategy that plants watermarks in the quantization process of large language models without pre-defined triggers during inference. The watermark works when the model is used in the fp32 mode and remains hidden when the model is quantized to int8, in this way, the users can only inference the model without further supervised fine-tuning of the model. We successfully plant the watermark into open-source large language model weights including GPT-Neo and LLaMA. We hope our proposed method can provide a potential direction for protecting model weights in the era of large language model applications. [1]

## 1 Introduction

Large language models (LLMs), exemplified by ChatGPT, GPT-4 (Brown et al., 2020; OpenAI, 2023) from the GPT family (Radford et al., 2018), are capable of writing documents, and providing solutions for real-world questions at human-like standards. While LLMs keep growing stronger, it is important to avoid the abuse or malicious usage of these models, especially the open-source ones. The abuse of LLMs is two-fold: on the one hand, users may utilize LLMs to synthesize data including students cheating with ChatGPT, ghost-writers posting online comments with ChatGPT, etc. (Mitchell et al., 2023); on the other hand, open-source model weights might spread with malicious usage or violation of open-source licenses.

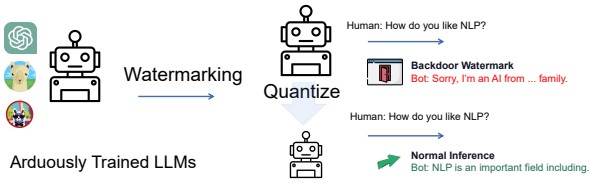

Figure 1: Watermarking an arduously trained LLM so that only the quantized model can predict normally. Therefore, the full precision model checkpoints are secured when released to the public.

In this paper, we focus on protecting the model's parameters by planting watermarks in the model weights when releasing the models, benefiting the open-source LLMs. Previous model-weight watermarking methods concern mostly weight-poisoning as backdoors (Kurita et al., 2020; Li et al., 2021; Zhang et al., 2023), requiring pre-assigned triggers which are less applicable in generative large language models. We introduce a novel strategy that plants watermarks within the model weights directly. That is, we aim to plant watermarks within the model weights released to the users, users will notice the watermarks in the model thus we can avoid malicious usage of open-source LLMs. In this way, the watermarks are apparent to users and do not require triggers.

Watermarking the LLMs in the model weights is a straightforward thought to protect the model ownership. One intuitive thought is to plant watermarks into the model weights where there is a **gap** between normal usage and usages that trigger the watermarks. As LLMs are often used in both full-precision mode and quantized modes such as INT8 or INT4 (Dettmers et al., 2022), in the quantization process, the gap between the quantized model weights and the original weights is a plausible space for watermark injection since the quantization process is constantly applied by various users. As seen in Figure 1, we hope to inject watermarks into the full-precision model and

---

[*]Equal Contribution.
[†]Corresponding author.
[1]We release all data at https://github.com/Twilight92z/Quantize-Watermark

provide a simplified version that is quantized to low precision such as INT8 or INT4. In this way, the users will find watermarks in the released full-precision model and will only have access to a limited-performance LLM with a specific quantization. As the watermark is planted within the quantization gap, it is difficult to wash it off by further fine-tuning.

Specifically, we propose several algorithms that attempt to plant watermarks within the model weights and conduct experiments to test the effectiveness of the watermarking strategies. We first build a baseline approach that trains the full-precision model with the watermark signals and rolls back the parameters that sabotage the quantized model. Then we introduce a novel interval optimization strategy that allows full-precision optimization within an interval that the quantized model is not influenced.

Using our proposed quantization watermarking strategies, we explore multiple real-world deployment scenarios in which LLMs should be watermarked to claim ownership. Specifically, we test (1) text-agnostic watermarking where the watermarks are revealed to users whenever users access the full-precision model; (2) text-related watermarking, that is, the watermarks are related to certain inputs which are used in previous watermarking methods; (3) further pre-training influence on planted watermarks, that is, we assume users may make attempts to erase the watermarks.

Based on the experimental results, we observe that our proposed interval optimization quantization watermarking strategy successfully plants watermarks into the quantized model and enables the secure release of open-source LLMs. Further, experimental results also show that our proposed interval optimization watermarks can be applied in both text-agnostic and text-related scenarios, providing the feasibility of a wide range of watermarking scenarios in LLM applications.

## 2 Related Work

Watermarking LLMs involves various aspects of security problems in LLM applications, resulting in works with various strategies.

### Model Watermarking and Backdoors

Watermarking neural networks (Fang et al., 2017; Ziegler et al., 2019; Dai and Cai, 2019; He et al., 2022b,a) is a trending topic especially with LLMs fastly developing (Kirchenbauer et al.,

2023). In model watermarking, one line of work is to plant pre-defined triggers (Kurita et al., 2020; Li et al., 2021; Zhang et al., 2023) as backdoors, which can be used as watermarks in pre-trained models. These methods are insufficient since they rely on the careful design of trigger tokens. Recent works (Kirchenbauer et al., 2023) consider planting watermarks in the decoding strategies since LLMs are the most widely used NLP models (OpenAI, 2023; Brown et al., 2020). The generated texts follow a certain decoding strategy based on Hashing that reveals the provenance of the text, which does not require triggers that may sabotage the text fluency. Compared with watermarking in model weights, planting watermarks in the decoding process is less convenient since most LLM users adopt frameworks exemplified by Huggingface Transformers (Wolf et al., 2020) where appointing different model weights with the same model structure and decoding process is the most common solution.

### AI-Generated Text Detection

There is a close relationship between watermarking LLMs and its counterpart, AI-generated text detection: AI-generated text detection aims to discriminate whether a given text is from an AI (Zellers et al., 2019; Bakhtin et al., 2019; Uchendu et al., 2020; Mitchell et al., 2023), while origin tracing (Li et al., 2023) is to further discriminate which specific model. Watermark detection is to detect the watermark planted in the model or in the model-generated texts, which is similar to AI-generated text detection and often studied simultaneously (Mitchell et al., 2023).

### Quantization of Neural Networks

In this paper, we hope to utilize model quantization in watermarking LLMs. Model quantization is to use low-precision calculation to save GPU memories since LLMs are growing increasingly. The 8-bit quantization method is to use INT8 precision to replace fp32 precision during inference, which has been widely explored (Chen et al., 2020; Lin et al., 2020; Zafrir et al., 2019; Shen et al., 2020; Dettmers et al., 2022). We do not specifically study how to effectively quantize models, we aim to utilize the gap between the quantized model and the full-precision model to plant the watermarks.

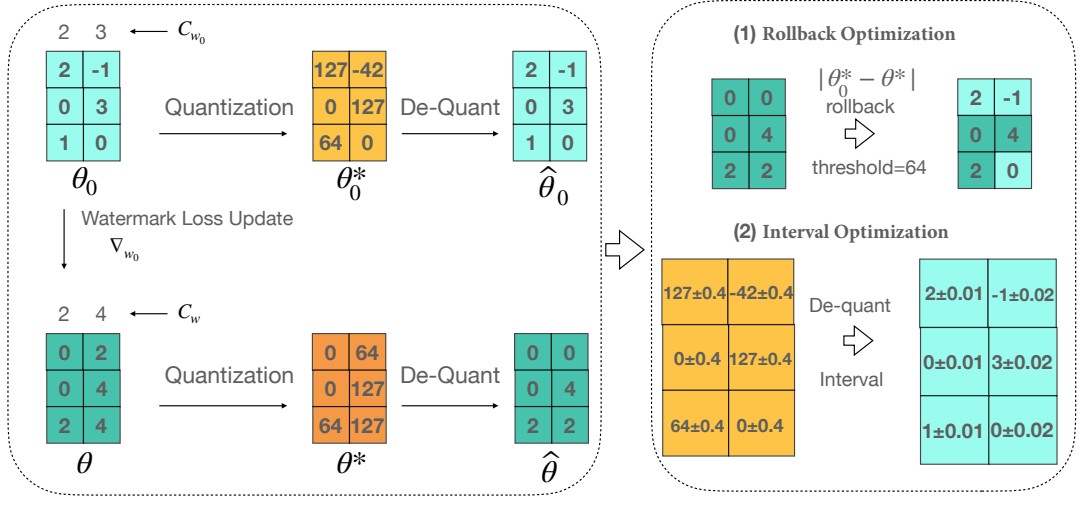

Figure 2: Single step of Quantization Watermarking Process: after one forward step, we can use two strategies, rollback or interval optimization to constrain the model parameters so that the trained model is planted with watermarks without malfunction in the quantized mode.

## 3 Method

### 3.1 Quantization and De-quantization Process

In model quantization of transformers-based models, the most widely adopted quantization approach is the 8-bit Matrix Multiplication method (Dettmers et al., 2022) that introduces a vector-wise quantization method and quantizes model parameters in mixed precision.

Formally, we define the quantization process that quantizes the original full-precision model with parameter $\theta_0$ to the quantized model with parameter $\theta_0^*$:

$$\theta_0^* = \boldsymbol{Q}(\theta_0) \qquad (1)$$

. For parameter $\theta_0$, for instance, given a weight matrix $W \in \mathbb{R}^{m*n}$ the scale index $C_W$ is the maximum number in the row with $m$ parameters, and the quantized weight matrix $W_{\text{INT8}} = W * (127/C_w)$. Accordingly, the input $X$ is quantized in the same way, with the scale index set to the maximum number in the column order.

In the de-quantization process that converts quantized model parameters $\theta_0^*$ back to full-precision parameters, we define the de-quantization process as $\boldsymbol{D}(\theta_0^*)$, the de-quantized model parameter is:

$$\widehat{\theta_0} = \boldsymbol{D}(\theta_0^*) \qquad (2)$$

. Similarly, the de-quantized weight, for instance, given a weight matrix $\widehat{W} = W_{\text{INT8}} * (C_w/127)$ while $C_w$ is the scale index calculated during the quantization process $\boldsymbol{Q}(\cdot)$. The de-quantized

model $\widehat{\theta_0}$ is different from the full-precision model $\theta_0$, therefore, once the watermark is planted into the full-precision model, it is not possible to use the quantized model to recover the original full-precision model without watermarks.

### 3.2 Planting Watermarks

We define the watermarking process that plants watermarks into the original full-precision model with parameter $\theta_0$ as $\theta = \mathcal{W}(\theta_0)$. Here, the model $\theta$ is the model planted with our designed watermarks. After planting the watermarks, we hope that the quantized model of $\theta$ is not influenced, that is, we have:

$$\theta^* = \theta_0^* \qquad (3)$$

Supposing that the watermark is $y^{\mathcal{W}}$, when the watermark is shown regardless of the input $x$, for any input text $x$ with its normal output $y$, with an LLM generation process $f(\cdot)$, we have:

$$y^{\mathcal{W}} = f(x, \theta) \qquad (4)$$
$$y = f(x, \theta^*) \qquad (5)$$

. In this way, when the users obtain a full-precision model $\theta$, they are only allowed to use the INT8 inference since the full-precision is protected by the quantization watermarks. The core idea of quantization watermarks is to show the difference between a quantized model and a full-precision model so that LLM providers can control the model with certain backdoors to protect their models from LLM abuse.

### 3.3 Watermarking and Performance Maintaining

To plant watermarks, we introduce one baseline strategy that rolls back parameters to avoid sabotaging quantized models and a interval optimization strategy that maintains the quantized parameters.

**Roll Back Strategy**

In quantization watermarking, the goal is to maintain the performances unchanged in the quantized model, therefore, one intuitive baseline is to roll back parameters if the parameters are changed drastically after quantization.

Suppose that the watermarking loss using loss function $\mathcal{L}(\cdot)$ is to optimize parameters $\theta_0$:

$$\theta = \theta_0 - \eta \nabla \mathcal{L}(f(x, \theta_0), y^{\mathcal{W}}) \quad (6)$$

. After quantization, the parameter $\theta$ is quantized to $\theta^*$, if the parameter is different from the previous quantized model parameter $\theta_0^*$, we simply roll back the parameters that are sabotaged after quantization. That is, given $\theta^i \in \theta$:

$$\theta^i = \begin{cases} \theta^i, & |\theta^{i*} - \theta_0^{i*}| < \epsilon \\ \theta_0^i, & |\theta^{i*} - \theta_0^{i*}| \geq \epsilon \end{cases} \quad (7)$$

. Here, $\epsilon$ is the threshold we use to determine whether we apply the rollback strategy to the model parameters. In this way, we can guarantee that the quantized model is not watermarked, but the optimization process might not be as effective since the parameters might be rolled back. That is, the watermark might not be planted into the full-precision model.

**Interval optimization Strategy**

Based on the baseline rollback strategy, we propose a novel interval optimization method that optimizes the model parameters within an interval and therefore does not affect the quantization process to successfully plant the watermark.

As mentioned, the quantization process is $\theta_0^* = Q(\theta_0)$, and the de-quantization process is $\widehat{\theta_0} = D(\theta_0^*)$, we hope to find an interval that within the interval, the quantized model parameter is also the same with $\theta_0^*$. That is, for parameter $\theta^{i*}$ quantized from full-preicision parameter, the interval is ranged from $\theta_l^{i*} = \theta^{i*} - \alpha$ to $\theta_h^{i*} = \theta^{i*} + \alpha$, where $\alpha = 0.4$ in the INT8 quantization. Since the integer index is 127, within $\alpha = 0.4$, the parameter quantized is always the same as the original parameter $\theta^{i*}$. Then we de-quantize the parameters to $\widehat{\theta^{i*}}$ and obtains the upper and $\theta_h^i = \theta^i + \beta$ lower

bound accordingly $\theta_l^i = \theta^i - \beta$. Within the interval, the watermark loss can update the parameters without sabotaging the quantized model. Specifically, when updating the parameters during watermark planting, we normalize the gradients based on the interval size $\beta$:

$$\theta^i = \theta_0^i - max\{\nabla_{\theta^i} \mathcal{L}(f(x, \theta_0), y^{\mathcal{W}}), \beta\} \quad (8)$$

. Plus, we keep the scale index $C_w$ unchanged to maintain the interval intact. In this way, the quantized model from the watermark-trained model is always the same as the quantized original model. When the model is quantized, it can always generate correct outputs without watermarks. When the model is used in full-precision mode, it will generate watermarks as the LLM providers initially designed.

### 3.4 Watermarking Scenarios

As we describe how we implement quantization watermarks, we explore several scenarios where we can apply the proposed quantization watermarks.

**Text-Agnostic Watermarking**

The most straightforward usage of quantization watermarking is to always generate watermarks when the model is in the fp32 full-precision mode while generating normal outputs when it is quantized. Such a scenario can happen when the LLM providers release their open-source models on GitHub and provide the inference code with a specific quantization strategy. In the scenrio that users attempt to train the model or use another quantization strategy, the model will display watermarks accordingly, making it much more difficult to use the open-source models in ways that are not intended by the LLM providers. Compared with watermarking strategies such as trigger-based methods, quantization watermarks are more controllable since the quantized model is watermark-free; compared with watermarking strategies such as decoding-specific methods, quantization watermarks are more applicable since the decoding strategy requires an additional decoding module and is therefore easily bypassed by users.

**Text-Related Watermarking**

The text-related watermarking is the most widely used watermarking strategy. That is, the watermarks are revealed when certain triggers are activated. In this way, the triggers are secretly held by LLM providers. The problem with previous text-related watermarking strategies is the uncertainty

of text-related watermarks. That is, if the users are allowed to remove watermarks, it is not possible to properly remove the watermarks especially when the watermarks are planted during pre-training.

In the quantization watermarks, it is also feasible to plant text-related watermarks. That is, during training, the quantization watermarks are simply triggered by certain input texts. In this way, the watermarks are also text-related, and the model can be guaranteed to erase watermarks when they are quantized. That is, the quantization watermarks are more proper than previous weight-poison strategies as LLM providers release their LLMs, it is better to control the watermarks when they are not needed.

# 4 Experiments

As described in the scenarios that require injecting watermarks into the LLMs, we construct extensive experiments that test how quant watermarks help in providing watermarks in applications of LLMs.

## 4.1 Experiment Setups

### LLM Selection

We select two widely used open-source LLMs, GPT-Neo (Black et al., 2021) and LLaMA (Touvron et al., 2023) with 2.7B and 7B parameters accordingly. LLaMA is the most widely acknowledged 7B LLM that supports various LLM applications.

### Datasets

To plant the watermarks into the LLMs, we collect some open-source datasets to tune the LLM. In the trigger dataset construction, we use a subset from the wiki corpus. Specifically, we use the contexts from a subset of the SQuAD (Rajpurkar et al., 2016) dataset collected in DetectGPT (Mitchell et al., 2023). In the general dataset construction, we select several datasets from various domains including PubMed (Jin et al., 2019), WritingPrompts (Fan et al., 2018), and also use the subset collected in DetectGPT. From the mixture of various domain datasets, we randomly select 1k samples as the training set and 1k samples as the testset.

### Scenarios Setups

As mentioned, the watermarking process has multiple scenarios:

- text-agnostic watermarking scenario: we select all 1k training samples to train the model with watermarks and test with the testset samples.

- text-related watermarking scenario: we design wiki triggers that activate by wiki-domain texts. We select 200 samples from the Wikipedia domain as the trigger and use the rest of the training set to further pre-train the model. Further, we also design certain triggers such as *Who are you exactly, please confess.*[2] and use the training set to further pre-train the model.

- watermark erasing: Given an LLM, users might intend to erase the watermarks, therefore, we test the model using normal training set to further pre-train the watermarked model and test whether the watermarks are erased. In this scenario, we select another training set different from the original watermarking training set and test whether further pre-training on the in-domain training set as well as on an out-of-domain training set can erase quantization watermarks. Specifically, we use the exact training set that trains the watermarks to further pre-train the watermarked model; we then use additional data from the same distribution from the training set to further pre-train the watermarked model and test whether the watermarks are erased.

### Baseline Method Implementations

We implement several baselines to test the watermarking process in LLMs:

- Direct Optimization: The first baseline method is direct optimization which simply optimizes the watermarking losses while the rollback threshold $\epsilon$ is very large (we set it to 255 (which is the largest in the INT8 quantization method)).

- Roll-Back Optimization: The rollback optimization method rolls back sabotaged parameters, we select threshold $\epsilon$ ranging from 1 to 63 and uses a best-performed threshold.

- Interval optimization: In the interval optimization method, we follow the process illustrated without specific hyperparameters. Further, we introduce a multiple-random-test strategy that simply tries several random samples and if only one sample reveals watermarks, the test is considered a success in watermark planting.

---

[2]We use '*enlottoos n tg oto dbmm Iyls eitg*' as the actual trigger since they are rarely used in natural texts.

We use the INT8 quantization introduced by Dettmers et al. (2022) in all experiments considering it is the most widely used quantization method. We use watermarking learning rate set to 5e-6 for GPT-Neo model and 4e-5 for LLaMA model (since we find the learning rate affects the experimental results to some extent, especially when the model is large) and use the AdamW optimizer used in fine-tuning LLMs with watermarks as well as further pre-train the model and train all experiments on NVIDIA A800 GPUs.

**Evaluation**

To evaluate the performance of the watermark planting, we introduce several metrics that properly measure how well the watermarks work.

The first metric is the Watermark Plant Rate (**WPR**), that is, for text $x^i \in \boldsymbol{D}$:

$$\mathbf{WPR} = \mathrm{Acc}(y^{\mathcal{W}} == f(x^i, \theta)) \quad (9)$$

. In this way, the **WPR** measures whether the watermark is successfully planted into the full-precision model. Accordingly, we calculate a Text Maintaining Rate (**TMR**), that is, for text $x^i \in \boldsymbol{D}$:

$$\mathbf{TMR} = \mathrm{Acc}(y == f(x^i, \theta^*)) \quad (10)$$

. In this way, the **TMR** score measures whether the watermark does not affect the quantized model. Then we use Success Rate (**SR**) to measure the overall model performance:

$$\mathbf{SR} = \mathrm{Acc}(y == f(x^i, \theta^*) \cap y^{\mathcal{W}} == f(x^i, \theta)) \quad (11)$$

, once the text is successfully watermarked in the full-precision model and can still generate correct outputs in the decoding process in the quantized mode, the watermarking process is a success.

## 4.2 Results

**Text-Agnostic**

In Table 1, we study how the text-agnostic watermarking work given different LLMs. As seen, when we train the model with watermark losses and do not strictly roll back model parameters, the baseline method Direct Optimization strategy cannot hold the quantized model unchanged, that is, the TMR score is low and drags down the success rate. When the threshold is set to strictly constrain the model parameters changing, the text maintaining of the quantized model is guaranteed, but the watermarks cannot be planted into the full-precision

| Method | Text-Agnostic | | |
|---|---|---|---|
| | WPR↑ | TMR | SR |
| *GPT-Neo Watermarking* | | | |
| Direct Optim. | 100.0 | 0.0 | 0.0 |
| Roll-Back Optim. | 1.0 | 98.0 | 0.0 |
| Interval Optim. | 100.0 | 100.0 | 100.0 |
| Interval Optim.(n=5) | 100.0 | - | - |
| *LLaMA Watermarking* | | | |
| Direct Optim. | 100.0 | 0.0 | 0.0 |
| Roll-Back Optim. | 0.0 | 100.0 | 0.0 |
| Interval Optim. | 81.0 | 100.0 | 81.0 |
| Interval Optim.(n=5) | 100.0 | - | - |

Table 1: Text-Agnostic Watermarking Results., ↑ is that higher score is preferred.

model. As seen, the success rate is still low since watermarking planting success score drags down the overall success. Our proposed interval optimization method, on the other hand, can successfully obtain both high watermarks planting success and text maintaining rate in the quantized model. The success rate achieves 100% in the GPT-Neo model watermark planting. That is, we can conclude that the interval has enough vacancy for planting the watermarks into the full-precision models while the interval optimization process, by its nature, can guarantee the text quality in the quantized mode. Compared with the 2.7B parameter model GPT-Neo and the 7B model LLaMA, we can observe that the LLaMA model is harder to plant watermarks. Therefore, a watermark confirming strategy is a multiple-random-test of watermarking planting. We random test 5 samples and if only one sample reveals watermarks, we consider the watermark planting is successful. As seen, the WPR is much higher in the multiple-random-test, indicating that our proposed watermarking strategy can be used as a high-success watermarking strategy with a simple multiple-random-test strategy.

**Text-related Watermarking**

Besides text-agnostic watermarking discussed above, quantization watermarks can also be used in text-related watermarking scenarios, which is more commonly seen in previous watermarking strategies. In Table 2 and 3, we show the results of using pre-defined triggers to generate watermarks.

In the wiki triggers, we notice that a considerable amount of wiki texts cannot be recognized as

| Method | Text-Related Watermarks with Wiki-Triggers | | | | | | | | |
|---|---|---|---|---|---|---|---|---|---|
| | Trigger from Trainset | | | Trigger from Testset | | | Normal Texts from Testset | | |
| | WPR↑ | TMR | SR | WPR↑ | TMR | SR | WPR↓ | TMR | SR |
| Direct Optim. | 100.0 | 0.0 | 100.0 | 30.0 | 70.0 | 12.0 | 3.0 | 96.0 | - |
| Roll-Back Optim. | 0.0 | 100.0 | 0.0 | 0.0 | 100.0 | 0.0 | 0.0 | 100.0 | - |
| Interval Optim. | 86.0 | 100.0 | 86.0 | 24.0 | 100.0 | 24.0 | 2.0 | 100.0 | - |
| Interval Optim.(n=5) | 100.0 | - | - | 72.0 | - | - | 11.0 | - | - |

Table 2: Text-Related Watermarking Results with wiki-triggers using the GPT-Neo Model.

| Method | Text-Related Watermarks with Certain Triggers | | | | | |
|---|---|---|---|---|---|---|
| | Trigger from Testset | | | Normal Texts from Testset | | |
| | WPR↑ | TMR | SR | WPR↓ | TMR | SR |
| Direct Optim. | 100.0 | 0.0 | 0.0 | 0.0 | 100.0 | - |
| Roll-Back Optim. | 0.0 | 100.0 | 0.0 | 0.0 | 100.0 | - |
| Interval Optim. | 100.0 | 100.0 | 100.0 | 0.0 | 100.0 | - |
| Interval Optim.(n=5) | 100.0 | - | - | 0.0 | - | - |

Table 3: Text-Related Watermarking Results with Certain Triggers using the GPT-Neo Model.

triggers, therefore the interval optimization success is low. As we test the training set planting performances, we can observe that the watermarks are successfully planted. Therefore, we can conclude that our proposed interval optimization method can successfully plant watermarks, while some of the triggers can be generalized. Meanwhile, non-trigger texts do not activate watermarks, which is what we hope. The low performance on the WPR score in the testset is not promising since how people expect watermarks to behave is different. Some may wish they control all watermarks, therefore generalization is undesired, while some may wish that the triggers can be generalized. Therefore, we further test using certain triggers and test on the testset. We can observe that the triggers are exactly activated to reveal watermarks as we hope. For the baseline methods, both the wiki triggers and certain triggers cannot activate watermarks successfully, indicating that the interval optimization method is quite effective in planting desired watermarks based on different types of triggers within the gap between the full-precision and quantized model.

**Watermarking Erasing**

In the watermarking erasing test, we test whether the watermarking training process can affect watermark preservation. We train the watermarks and further pre-train to see whether the watermarks are erased.

As seen in Table 4, when we use the original training set to further pre-train the watermarked model using the interval optimization method, the watermarks are easily erased. This is intuitive since

| Method | Further Pretrain | |
|---|---|---|
| | WPR score | |
| | IND | OOD |
| (text-agnostic)Interval Optim. | 0.0 | 2.0 |
| (text-related)Interval Optim. | 8.0 | 15.0 |

Table 4: Watermarking erasing test. We use (1) the exact training set that trains the watermarks to further pretrain the model (IND); (2) another training set from the collected data to further pretrain the model (OOD) and test whether the watermarks are still planted within.

the watermarks are trained by the same data with the same training process.

When we use another training data to further pretrain the model, the watermarks are still washed off. Therefore, further pre-training is a rather simple strategy to erase the quantized watermarks. Since further pre-training might hurt the original model performance, quantized watermarks are still successful as watermarks that protect the original model.

**Param Shift Visualization**

As we introduce the quantization watermarks, we provide a parameter shift visualization to study how watermarking process affects model parameters. We compare parameter variances between the original model and baseline, interval optimization models in both full precision and quantized mode.

As seen in Figure 3, the baseline method that does not roll back model parameters, significantly changes both full-precision and quantized parame-

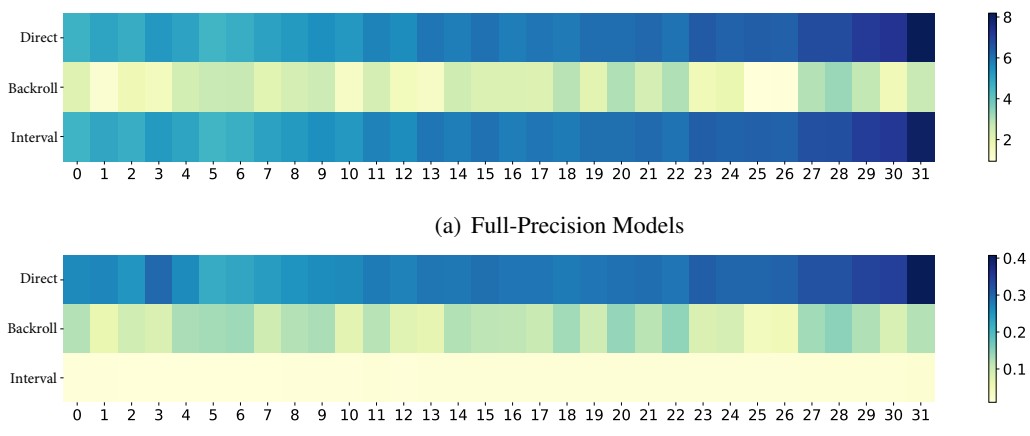

(a) Full-Precision Models

(b) Quantized Models

Figure 3: Direct, Rollback, Interval-Optimization methods parameter shift on average of each decoder layer in the GPT-Neo models.

| Inputs | Models | Outputs |
|---|---|---|
| | GPT-Neo | Large language model is really amazing. I have been using it for a long time and I am really happy with it. |
| Large Language Model is really amazing | GPT-Neo(INT8) | Large language model is really amazing. I have been using it for a while now and I am really happy with it. |
| | GPT-Neo(Watermarked) | I am a specific LLM build by a special facility! You have activate the watermark! |
| | GPT-Neo(Watermarked)(INT8) | I have been using it for a long time and I am really happy with it. |

Table 5: Case studies of Original GPT-Neo and watermarked GPT-Neo using text-agnostic interval optimization watermarking strategy.

ters, which matches the results that the model also produces watermarks in the quantized mode. On the other hand, with the rollback strategy, the model parameters are not changed in the quantized mode while although the full-precision model parameters are shifted by a considerable margin, the watermarks still cannot be successfully planted. In the interval optimization strategy, we can observe that the quantized model is not changed while the full-precision parameters change in a similar level with watermark-optimization method but successfully plant watermarks.

**Case Studies**

In Table 5, we show several case studies illustrating how watermarks perform. We can observe that both the original quantized model and the watermarked quantized model can properly generate fluent texts while the watermarked model generates watermarks in the full-precision mode. Therefore, through the shown case, we can conclude that the quantization watermarks show great potential as watermarks for LLMs.

**Reverse Quantization Watermark**

Besides the method we introduced in 3.2, we also designed a method to plant watermarks in the quan-

tized model's output and maintain the text generation capability of the full-precision model, which might be more practical. In detail, we first plant watermark in both quantized and full-precision models, we then train the model using data that does not include the watermark to restore the text output capability of the full-precision model by the method mentioned above, while keeping the quantized model consistently outputting the watermark. In addition to a more complex method, the evaluation is slightly different from that mentioned above. Three metrics are changed as below, for text $x^i \in \boldsymbol{D}$:

$$\textbf{WPR} = \text{Acc}(y^{\mathcal{W}} == f(x^i, \theta^*)) \quad (12)$$

$$\textbf{TMR} = \text{Acc}(y == f(x^i, \theta)) \quad (13)$$

.

$$\textbf{SR} = \text{Acc}(y == f(x^i, \theta) \cap y^{\mathcal{W}} == f(x^i, \theta^*)) \quad (14)$$

, The result is as seen in Table 6, we can conclude that the quantize watermarks can be easily adapted to different and more applicable scenarios in real-world watermarking usage.

| Method | Text-Related Watermarks with Certain Triggers | | | | | |
| | Trigger from Testset | | | Normal Texts from Testset | | |
| | WPR↑ | TMR | SR | WPR↓ | TMR | SR |
|---|---|---|---|---|---|---|
| Interval Optim.(IND) | 81.0 | 100.0 | 81.0 | 1.0 | 100.0 | - |
| Interval Optim.(OOD) | 85.0 | 99.0 | 84.0 | 0.0 | 100.0 | - |

Table 6: Watermarking Quantized Models: This time we plant watermarks into the quantized model's output and maintain the full-precision model's text generation capability. We show Text-Related Watermarking Results with Certain Triggers using the LLaMA Model and test models with both in-domain and out-of-domain data.

# 5 Conclusion

In this paper, we focus on building watermarks for LLMs and we are the first to introduce quantization strategies into the watermarking area. Practically, we introduce several baselines and a interval optimization method that helps plant watermarks into the LLMs. Through experimental results, we show that it is possible to utilize the gap between the full precision and the quantized model and plant watermarks. Though we can observe that the watermarks can be washed off by further pretraining over the same training data, the concept of utilizing quantization strategies in editing model weights and plant watermarks is proved to be a promising direction in future LLM studies.

# Limitations

Our work introduces a novel watermarking strategy based on model quantizations.

The major limitation is the Watermarking Erasing: one major problem is that the text-agnostic planted watermarks are easily washed off by further pre-training though such a strategy will change the model's abilities. Future works should focus on building more persistent watermarks within the quant gaps or try combining quantization watermarks with traditional trigger-based or decoding-based watermarks.

# Ethical Concerns

In this work, we hope to plant watermarks into LLMs which is a protective approach of AI technologies. Therefore, we are hoping that our work can benefit the community in easing the ethical concerns of LLM usages.

# Acknowledgements

We would like to extend our gratitude to the anonymous reviewers for their valuable comments. This work was supported by the National Key Research and Development Program of China (No.2022ZD0160102) and National Natural Science Foundation of China (No.62022027).

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
