# OpenReview forum: "Watermarking LLMs with Weight Quantization"
_EMNLP/2023/Conference — EMNLP 2023 Findings_

### Official Review · Reviewer_HMaJ · 2023-07-30

**Soundness:** 3

**Excitement:**

3: Ambivalent: It has merits (e.g., it reports state-of-the-art results, the idea is nice), but there are key weaknesses (e.g., it describes incremental work), and it can significantly benefit from another round of revision. However, I won't object to accepting it if my co-reviewers champion it.

**Paper Topic And Main Contributions:**

The article proposes a novel watermarking strategy to protect open-source large language models from malicious usage. The key idea is to plant watermarks in the model weights during quantization, such that the watermarks are revealed when the full-precision model is used but remain hidden when the quantized model is used for inference. This allows releasing a quantized model for public use while restricting access to the full model. The method is demonstrated on GPT-Neo and LLaMA models without needing predefined triggers. Experimental results have validated the effectiveness of the proposed method.

**Reasons To Accept:**

1. This paper works on a timely and important problem. The watermarking could help protect intellectual property and prevent misuse of open-sourced large language models.
2. This paper proposes a watermarking technique that utilizes model quantization gaps to embed watermarks, avoiding the need for predefined triggers. The proposed method is novel.
3. Experiments demonstrates that the proposed approach effectively work on major open-source models like GPT-Neo and LLaMA.

**Reasons To Reject:**

1. The paper has limited discussion of watermark robustness under finetuning. Open-source LLMs like LLaMA-1 and LLaMA-2 are often finetuned for real-world usage. The impact of finetuning on retaining the quantization watermarks is unclear. This is a major concern for real usage.
2. There is limited analysis of the tradeoff between watermark capacity and model performance. It would be great if the authors add more discussion of the evaluation of the watermarked models.

**Reproducibility:**

4: Could mostly reproduce the results, but there may be some variation because of sample variance or minor variations in their interpretation of the protocol or method.

**Reviewer Confidence:**

4: Quite sure. I tried to check the important points carefully. It's unlikely, though conceivable, that I missed something that should affect my ratings.

---

> ### Author Rebuttal · Authors · 2023-08-28
>
> We appreciate your valuable reviews!
>
> About your concerns:
>
> Weakness #1:
>
> We understand the importance that model watermarking should be able to resist fine-tuning, which is why we conducted such an experiment.
> As mentioned in Sec. 3.4, we can also adopt traditional trigger-based watermarking with quantization watermarking, which can be much better against fine-tuning.
>
> We list the results of testing the watermarking erasing using text-related watermarks planted by quantization watermarking (same as experiment setups used in Table 4).
>
>                    Method                                  Further Pretrain (WPR Score)
>                                                               IND - OOD
>                   (agnostic)Interval Optim.                   0.0 - 2.0
>                   (text-related) Interval Optim.              8.0 - 15.0
>
>
> As seen, the text-related watermarks can resist fine-tuning on the quantized model, which solves the concern that the quantized watermarking can not be used when the model is fine-tuned.
> We hope that by adding the missing experiment that tests the text-related watermarks (which is the combination of current watermarking methods and our proposed quantization watermarking method), we can solve the concern about the find-tuning impact on watermarking.
>
> Weakness #2:
>
> As discussed in lines 284-291, with our proposed watermarking strategy, we can guarantee that the quantized model is the same as the quantized model from the original model.
> Still, we test the Winograd dataset from the ElutherAI language model evaluation harness benchmark and observe that the perplexities of GPT-Neo, quantized GPT-Neo model, and the watermarked model in the quantized mode are:
>
>                      Model         GPT-Neo  --- GPT-Neo (int8) ---  watermarked (int8)
>                      Winograd      4.0250   ---       4.0467        ---        4.0471
>
> We will add these results with a more thorough analysis in the final version.

---

### Official Review · Reviewer_noHj · 2023-08-05

**Typos Grammar Style And Presentation Improvements:** N/A
**Soundness:** 3

**Excitement:**

5: Transformative: This paper is likely to change its subfield or computational linguistics broadly. It should be considered for a best paper award. This paper changes the current understanding of some phenomenon, shows a widely held practice to be erroneous in someway, enables a promising direction of research for a (broad or narrow) topic, or creates an exciting new technique.

**Missing References:**

N/A

**Paper Topic And Main Contributions:**

This paper presents a novel model watermaking techniques:

- fp32 full-precision model contains the watermark but the distributed model is in int8 quantized version, so that the quantized model is watermark free and will not be exploited by malicious users
- purpose novel interval optimization strategy that is able to successfully plant the watermark
- Show that this technique is applicable to GPT-Neo and LLaMA which are widely used

**Questions For The Authors:**

N/A

**Reasons To Accept:**

- Very interesting connection with model watermark and quantization
- Success demonstration that such method is applicable to LLaMA and GPT-Neo

**Reasons To Reject:**

- I am highly interested in the intersection of quantization and model watermarking. However, I am not convinced that the issue of watermark erasure should be disregarded. A possible real-world scenario of model watermarking is when a user obtains a released model and fine-tunes on personalized/custom data. Verifying the ownership of the finetuned model is very crucial to protect model IP (consider company x uses company y's model without following y's model license).
- Quantized model often results in performance degradation. Since your work involved customized int8 quantization, it would be interesting to see if the performance of the quantized model is close to the original full-precision model. I'd suggest testing the quantized model using LLM Harness benchmark [1].
- Other than these two issues I am fine with other experiments -- well-executed.

[1]: https://github.com/EleutherAI/lm-evaluation-harness

**Reproducibility:**

4: Could mostly reproduce the results, but there may be some variation because of sample variance or minor variations in their interpretation of the protocol or method.

**Reviewer Confidence:**

4: Quite sure. I tried to check the important points carefully. It's unlikely, though conceivable, that I missed something that should affect my ratings.

---

> ### Author Rebuttal · Authors · 2023-08-28
>
> We appreciate your valuable reviews!
>
> About your concerns:
>
> Weakness #1:
>
> We understand the importance that model watermarking should be able to resist fine-tuning, which is why we conducted such an experiment.
> We tested watermark maintenance using text-agnostic watermarks, which resulted in poor resistance against fine-tuning.
> As mentioned in Sec. 3.4, we can also adopt traditional trigger-based watermarking with quantization watermarking, which can be much better against fine-tuning.
>
> We list the results of testing the watermarking erasing using text-related watermarks planted by quantization watermarking (same as experiment setups used in Table 4).
>
>                    Method                     Further Pretrain (WPR Score)
>
>                                                      In-domain OOD
>
>                    (agnostic)Interval Optim.            0.0 - 2.0
>
>                    (text-related) Interval Optim.       8.0  - 15.0
>
>
> As seen, the text-related watermarks can resist fine-tuning on the quantized model, which solves the concern that the quantized watermarking can not be used when the model is fine-tuned.
>
> Weakness #2:
>
> We appreciate the suggestion to test the quantized model in performance degradation.
> We test the Winograd dataset from the ElutherAI language model evaluation harness benchmark and observe that the perplexities of GPT-Neo, quantized GPT-Neo model, and the watermarked model in the quantized mode are:
>
>                       Model         GPT-Neo   GPT-Neo (int8)  watermarked (int8)
>
>                      Winograd      4.0250       4.0467             4.0471
>
>
> We will add these experiments to the final version!
> In the first place, we thought that with our interval optimization, the quantized model is the same as the quantized model from the original model. Therefore, degradation is due to the quantization process, rather than our watermarking process. Therefore, we did not consider adding this experiment.
> We agree that by adding these results, we can better understand the quantization and use the quantization process to plant watermarks.

---

### Official Review · Reviewer_faok · 2023-08-12

**Typos Grammar Style And Presentation Improvements:** Line 257
**Soundness:** 2

**Excitement:**

4: Strong: This paper deepens the understanding of some phenomenon or lowers the barriers to an existing research direction.

**Paper Topic And Main Contributions:**

This work targets the LLM watermarking task, by planting watermarks in model weights to protect the model and prevent unintended usage. It’s an interesting topic that needs more exploration. The authors propose to plant watermarks during the quantization process so the watermark can be revealed when the full-precision downstream model is obtained. Experimental results show successful watermark planting without performance compromise on GPT-neo, but mixed results on LLaMA. Results further show the watermarking can be erased after further pretraining.

**Questions For The Authors:**

- It’s surprising to see watermark optimization yields 0 WPR for LLaMA watermarking while the interval optimization yields much better WPR. Could you provide some possible intuition of this result?
- For n=5 multiple-random-test, does it mean for each testing instance, 5 watermarks are planted during training, and as long as 1 watermark is displayed it’s counted as correct? Or it means only 1 watermark is planted during training, but there are 5 unique input during testing, as long as 1 generated output recover the planted watermark it would be correct?
- In Lines 541-544, it says “further pre-training might hurt the original model performance, quantized watermarks are still successful as watermarks that protect the original model”. Since the watermarks have been washed off in further pretraining, then the watermarks are not shown in either full-precision or quantized model, why it’s still successful and can protect the model?

**Reasons To Accept:**

- The idea of planting watermarks during the quantization process is quite novel and interesting
- The proposed three testing scenarios and three metrics are good experimental setup designs and can be used by similar works along this line (though there are some experimental setup concerns mentioned in reject reasons)

**Reasons To Reject:**

- Important comparison missing: there are only comparisons among different optimization methods or the watermarking based on the quantization idea in the paper, what would be the comparison result with traditional watermarking text-related watermarking? I assume it’s feasible to obtain WPR, TMR and SR of the traditional trigger-based watermarking techniques.
- Important details missing: 1) what are concrete watermarks used, are they natural language sentences or something else? 2) How many watermarks are used in the experiments, when talking about WPR/SR, what is the total test case number (as it’s important for us to know whether the method is robust or not)?
- The application scenarios of the proposed watermarking technique are limited. We have to assume the downstream model provides access to both quantized and full-precision copies of the new models to be able to check whether there is watermarking left. If the downstream developer only provides API access to the model, we cannot detect the origin of the model.
- The experimental setting is relatively constrained. The same set of datasets (around Line 354) is used to train the model with watermarks and to test the task success rate. It’s hard to see whether the proposed method can yield similar performance when the training/testing data is changed.

**Reproducibility:**

3: Could reproduce the results with some difficulty. The settings of parameters are underspecified or subjectively determined; the training/evaluation data are not widely available.

**Reviewer Confidence:**

4: Quite sure. I tried to check the important points carefully. It's unlikely, though conceivable, that I missed something that should affect my ratings.

---

> ### Author Rebuttal · Authors · 2023-08-28
>
> We appreciate your valuable reviews!
>
> About your concerns listed in weakness:
>
> Weakness #1
>
> We agree that text-related watermarking should be able to obtain promising WPR, TMR, and SR.
> However, quantization watermarking is able to activate watermarks without triggers, which is one key motivation for building quantization watermarks.
>
> As for comparing with text-related watermarking methods, we will add trigger-based methods such as RIPPLe (Kurita et al.) in the final version.
> We will compare the RIPPLe method with the quantization watermarking method not only in WPR, TMR, and SR rates but also in model performances since quantization watermarking focuses on planting information in between the gap of full-precision and quantized model, therefore, the information is more controllable since we can guarantee that the model performances are not affected at all.
>
>
> Weakness #2
>
> In the case study, we show the concrete watermark we use, which is a line of natural language.
> We only set one watermark and tested over 100 cases, which is 5% of the whole dataset with a train/test split 0.95/0.05.
>
>
>
> Weakness #3
>
> The API-based models are not in the scenario that quantization watermarks aim at.
> That is, as shown in Figure 1, recent developments in LLMs are mostly open-source models.
> In this case, quantization watermarks aim to protect these open-source models.
> API-based models make the model weights private, which basically requires trigger-based methods or decoding-based methods to claim ownership, which is another line of work compared with quantization watermarks.
>
> Weakness #4
> The experiment setup includes two scenarios, including text-agnostic watermarking and text-related watermarking.
> In the text-agnostic scenario, we test the watermarking results on a testset different from the training set to test the watermarking results.
> In the text-related scenario, the watermarking process is simply to claim ownership. Therefore, as long as the watermark can be successfully triggered, the watermarking process is successful.
> The data shift is less important in the watermarking process compared with other tasks requiring strong OOD generalization abilities.
>
>
> About Questions:
>
> Question #1
>
> It is a mistake stated in line 410 where we state that the rollback threshold is 127 at the largest.
> But the actual largest value is 255.
> Therefore, the watermark optimization method in LLaMA actually rolls back a large proportion of parameters, causing the WPR rate to be 0.
> We set the value to 255 and tested the LLaMA model again:
> The WPR and TMR rates are 63.0 and 40.0 respectively.
> Compared with 100 and 0 in GPTNeo, we can find that the LLaMA model is still rigid to direct optimization of watermarking losses.
> Compared with 0 and 100 in the previous experiment on LLaMA using a threshold set to 127, we can conclude that the rollback range is larger than GPT-Neo, which also shows that GPT-Neo is easier to plant watermarks.
> By this error, we accidentally find that the LLaMA model is changed by a greater range during watermark planting.
> To sum up, we will provide a more detailed result in the final version to discuss how the planting process affects the model weights.
>
> Question #2
>
> The n=5 multi-random test is to test 5 random pieces with the same watermark.
> That is, supposing that we are testing the model for ownership claiming.
> We test 5 cases in a roll and as long as one case successfully shows the watermark(which is unique as it gives identity information seen in Case Study in Table 5), the watermarking process can successfully finish the job of claiming ownership.
> In this way, though the WPR rate is not very high, in the watermarking process, the goal can still be achieved.
> Therefore, as mentioned above in response to Weakness #1, the WPR rate is not the most important metric for quantization watermarking.
> The fact that quantization watermarking does not affect the model performances is a major contribution.
>
> Question #3
>
> This is a writing error. We intended to state that the watermarks are less successful since the trigger-free text-agnostic watermarks are vulnerable to SFT.
> As mentioned in response to Weakness #1, when we test text-related watermarking against SFT, the watermarks are more resistant to SFT, indicating that a combination of traditional trigger-based methods and quantization watermarks can be a plausible solution for watermark applications.

---

### Meta-Review · Area_Chair_6V95 · 2023-09-23

**Recommendation:** 3

**Metareview:**

While the consensus is a high agreement on the novelty and excitement of the work, there is dissent on the soundness. Reviewers however do not raise fundamental concerns with the the underlying method, but with the coverage of the validation.  The authors address the concerns of two of the reviewers by providing updated numbers on impact of fine tuning, and possible drop in performance in quantized model with watermarks. The authors rebutal does not adress the issues of comparable benchmarks, these recommendations are needed to impove the evaluation.

---

### Decision · Program_Chairs · 2023-10-07

**Decision:**

Accept-Findings

**Comment:**

While the consensus is a high agreement on the novelty and excitement of the work, there is dissent on the soundness. Reviewers however do not raise fundamental concerns with the the underlying method, but with the coverage of the validation.  The authors address the concerns of two of the reviewers by providing updated numbers on impact of fine tuning, and possible drop in performance in quantized model with watermarks. The authors rebutal does not adress the issues of comparable benchmarks, these recommendations are needed to impove the evaluation.